# Aeolus: A Multi-structural Flight Delay Dataset

**Lin Xu**[1]* **Xinyun Yuan**[1]* **Yuxuan Liang**[2] **Suwan Yin**[1] **Yuankai Wu**[1]✉

[1]Sichuan University [2]The Hong Kong University of Science and Technology (Guangzhou)
xulin12138@stu.scu.edu.cn, wuyk0@scu.edu.cn

## Abstract

We introduce **Aeolus**, a large-scale *Multi-modal Flight Delay Dataset* designed to advance research on flight delay prediction and support the development of foundation models for tabular data. Existing datasets in this domain are typically limited to flat tabular structures and fail to capture the spatiotemporal dynamics inherent in delay propagation. Aeolus addresses this limitation by providing three aligned modalities: (i) a tabular dataset with rich operational, meteorological, and airport-level features for over 50 million flights; (ii) a flight chain module that models delay propagation along sequential flight legs, capturing upstream and downstream dependencies; and (iii) a flight network graph that encodes shared aircraft, crew, and airport resource connections, enabling cross-flight relational reasoning. The dataset is carefully constructed with temporal splits, comprehensive features, and strict leakage prevention to support realistic and reproducible machine learning evaluation. Aeolus supports a broad range of tasks, including regression, classification, temporal structure modeling, and graph learning, serving as a unified benchmark across tabular, sequential, and graph modalities. We release baseline experiments and preprocessing tools to facilitate adoption. Aeolus fills a key gap for both domain-specific modeling and general-purpose structured data research.Our source code and data can be accessed at `https://github.com/Flnny/Delay-data`

## 1 Introduction

Advances in machine learning have fueled rapid progress in real-world applications, yet translating academic success to industrial deployment remains nontrivial—especially in the domain of tabular data, where benchmarks often fall short in representing practical complexities [3]. In industrial settings, tabular datasets typically exhibit temporal distribution drift and contain a mix of predictive, redundant, and correlated features—products of elaborate data engineering pipelines [14]. However, academic tabular benchmarks largely ignore these factors: timestamp metadata is often missing, and feature sets are simplified, limiting the generalization of research findings [29]. Moreover, many applications involve not just static tabular features, but **multimodal signals**, such as sequences and graphs, reflecting complex real-world interactions [2]. Bridging this gap requires benchmarks that reflect these characteristics.

One such high-stakes domain is **flight delay prediction**, where economic losses, passenger disruptions, and carbon emissions are compounded by the failure to anticipate delay cascades [4]. While delays often originate from stochastic disruptions (e.g., weather or ATC interventions), their propagation follows intricate **spatiotemporal and relational dynamics** [44]. For example, a delayed flight may block downstream gate usage, affect crew rotations, or trigger airspace congestion—phenomena that tabular models alone cannot fully capture [43].

Despite the critical role of flight delay prediction in air traffic management, existing public datasets exhibit several limitations that hinder the development of realistic and generalizable models [39].

---

*These authors contributed equally.✉ Corresponding author

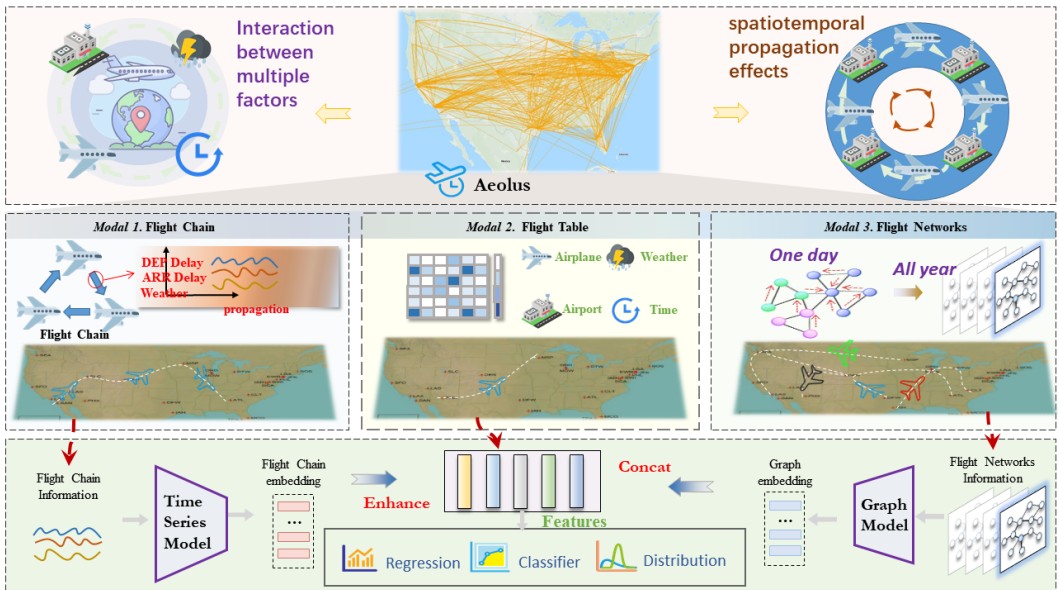

Figure 1: Overview of Aeolus.

Most widely used benchmarks consist solely of flat, flight-level tabular features—such as basic schedule information—while lacking richer modalities like aircraft rotation sequences or dynamic airport resource graphs [28]. Many datasets are geographically constrained (e.g., Nanjing Lukou Airport, U.S. domestic flights from 1995–2008), limiting their applicability to network-wide delay propagation across heterogeneous airports [29]. Several suffer from temporal leakage (e.g., including future weather data) or lack standardized multi-task evaluation protocols, which compromises reproducibility and benchmarking reliability [5]. Additionally, many datasets are outdated or biased toward major hubs, overlooking evolving operational patterns and the dynamics of smaller or regional airports [14]. Finally, datasets used in many published studies are not publicly accessible, further impeding transparency and community-driven progress [2].

To address both the *general* challenges of real-world tabular learning and the *specific* complexity of delay forecasting, we introduce **Aeolus**—a unified benchmark combining **tabular**, **temporal**, and **graph-structured** data. Aeolus captures:

- **Multi-granular Features** at flight-, airline-, and environment-levels.

- **Spatiotemporal Modeling** via aircraft-chain sequence and dynamic airport graphs.

- **Robust Evaluation Protocols** using temporal slicing and multi-task settings.

As shown in Figure 1, to address the complex interplay of spatiotemporal propagation and multi-factor interactions in flight delays, **Aelous** extract three aligned modalities from the raw flight operation and weather datasets: (1) Tabular data, consisting of flight-level structured features (e.g., schedule, weather, airport metadata) capturing static and contextual delay factors; (2) Flight chains, which model delay propagation within individual aircraft by sequentially linking flights operated by the same tail number within a 24-hour window, enabling temporal reasoning about upstream dependencies; and (3) Flight network graphs, which capture inter-aircraft delay interactions through shared resources (e.g., airport gates, airspace, crew), allowing relational modeling across different flights. These three modalities are jointly constructed and preprocessed to support three key delay prediction tasks—regression, classification, and uncertainty estimation—under a unified benchmark protocol. This design enables the study of delay dynamics from tabular, sequential, and graph-structured perspectives. Through Aeolus, we aim not only to improve delay prediction, but also to establish a testbed for studying tabular ML models under multimodal, temporally-evolving, and structurally-rich conditions—reflecting **broader industrial realities** beyond aviation.

## 2 Background

This section outlines the challenges of flight delay prediction, the limitations of existing datasets, and the gaps in current tabular learning benchmarks, underscoring the need for a more comprehensive and realistic benchmark for both flight delay prediction and tabular learning.

### 2.1 Challenges in Flight Delay Prediction

Flight delay prediction presents significant challenges for machine learning research due to the complex, dynamic, and interdependent nature of air transportation systems. Industrial datasets in this domain often exhibit **temporal distribution drift** caused by seasonal trends, policy changes, and operational adjustments [7]. Moreover, these datasets encompass a mixture of predictive, redundant, and correlated features resulting from extensive data engineering pipelines, necessitating robust preprocessing and management techniques to ensure model generalization.

As highlighted by Sternberg et al. [34], flight delay data spans multiple modalities: **tabular attributes** (e.g., scheduled departure times, flight numbers), **temporal sequences** (e.g., aircraft schedules), and **graph-structured relationships** (e.g., airport networks and airspace congestion). Integrating these modalities is critical for capturing delay dynamics but introduces substantial preprocessing and modeling hurdles. Recent studies have explored advanced machine learning techniques, such as graph neural networks, to model these complex relationships [8, 40].

The air transportation system is further complicated by **spatiotemporal and relational dynamics**. Delays propagate through operational dependencies—such as shared aircraft, crew schedules, and airport resource constraints—creating cascading effects that challenge traditional tabular approaches. For instance, a delayed inbound flight can disrupt gate assignments or crew availability, amplifying delays network-wide [40]. Modeling these dependencies requires joint consideration of temporal sequences and structural relationships, a task made difficult by the **non-i.i.d. nature** of flight data, where observations are interlinked through operational constraints.

The stakes of accurate prediction are high, with significant economic and environmental implications. Delays incur costs for airlines (e.g., fuel, penalties), airports (e.g., capacity bottlenecks), and passengers (e.g., missed connections), while also increasing carbon emissions from inefficient operations. For example, in 2022, flight disruptions in the U.S. alone led to economic losses estimated between $30–34 billion and generated approximately 3 million tons of additional $CO_2$ emissions [1]. These factors underscore the need for datasets and benchmarks that reflect real-world complexities, enabling models to deliver actionable insights for proactive delay mitigation.

### 2.2 Limitations of Existing Flight Delay Datasets

Despite the growing interest in flight delay prediction, publicly available datasets suffer from significant limitations that hinder the development of robust, generalizable models, as shown in the Table 1. We identify five key deficiencies:

**Spatiotemporal Limitations.** Most datasets span only short time periods (typically 1–5 years) and cover fewer than 100 airports [2, 32], with some focusing on a single location [26, 42]. They lack global coverage and long-term continuity, restricting their utility for studying delay patterns at scale.

**Missing Operational Signals.** Existing datasets rarely include critical multimodal features such as aircraft rotations, gate assignments, crew schedules, or passenger connections [34]. The absence of these operational variables limits feature richness and reduces model fidelity.

**Task Inflexibility.** Nearly 70% of flight delay datasets support only a single task—either regression or classification [36, 37]. None support uncertainty quantification, which is essential for risk-aware decision-making in operational environments.

**Lack of Unified Multitask Support.** No existing dataset supports all three key predictive tasks: delay duration estimation, delay occurrence classification, and uncertainty modeling. This hinders the development and fair evaluation of multitask learning approaches.

**Limited Accessibility.** Many datasets are behind access restrictions or licensing barriers, limiting reproducibility and slowing research progress. Open and fully accessible benchmarks remain rare.

These limitations motivate the need for a comprehensive benchmark that offers long-term, large-scale, and multimodal flight data with full support for multitask learning and uncertainty estimation.

Table 1: Comparison of Current Flight Delay Datasets.

| Dataset | Timeframe | Airports | Flights | Ext. Feat. | Mult. Mod. | Mult. Task. | Pub. Avail. |
|---|---|---|---|---|---|---|---|
| [2] | 5Y | 75 | 27.08M | ✓ | ✓ | ✗ | ✗ |
| [32] | 4Y | 373 | 12.34M | ✗ | ✓ | ✗ | ✗ |
| [36] | 3Y | 2 | 1,058 | ✓ | ✓ | ✓ | ✗ |
| [20] | 1Y | 58 | 5.42M | ✓ | ✗ | ✗ | ✗ |
| [37] | 1Y | 34 | 5.58M | ✗ | ✓ | ✓ | ✗ |
| [17] | 1Y | 10 | 5.7M | ✓ | ✓ | ✗ | ✗ |
| [22] | 1Y | 5 | 55.82M | ✓ | ✗ | ✗ | ✗ |
| [42] | 1Y | 1 | 10.15M | ✗ | ✗ | ✗ | ✗ |
| [23] | 8M | 366 | 5.51M | ✓ | ✗ | ✓ | ✗ |
| [26] | 3M | 1 | 21,298 | ✗ | ✓ | ✗ | ✗ |
| [21] | 1M | 348 | 0.63M | ✗ | ✓ | ✗ | ✗ |
| Aeolus(Ours) | 9Y | 320 | 54.67M | ✓ | ✓ | ✓ | ✓ |

## 2.3 Limitations of Tabular Datasets

We further find that our dataset is not only valuable for flight delay prediction but also meaningful for general tabular learning tasks. Table 2 highlights several key limitations in current tabular data benchmarks that impede the development of models with strong generalization capabilities in real-world applications:

**Temporal Leakage Due to Random Splits.** Many benchmarks employ random train-test splits, disregarding the temporal dependencies inherent in time-series data. This practice can lead to temporal leakage, where models inadvertently access future information during training, resulting in inflated performance metrics that do not reflect real-world scenarios [13, 24].

**Data Leakage and Synthetic Data Limitations.** Some datasets include features like user identifiers, which can artificially boost model performance. Additionally, reliance on synthetic datasets, such as the Artificial-Characters dataset, may fail to capture the complexity and variability of real-world data, leading to models that perform well in controlled settings but poorly in practical applications.

**Limited Domain Diversity and Sample Sizes.** Existing benchmarks predominantly focus on domains like finance and healthcare, with limited representation of complex areas such as aviation delay prediction. While TabReD [30] has expanded domain coverage, it still lacks the spatiotemporal dependencies and high-dimensional features characteristic of the aviation sector. Moreover, many datasets have relatively small sample sizes, restricting the development and evaluation of models intended for large-scale applications.

Aeolus complements TabRed by covering the unique characteristics of the aviation domain, facilitating the development of models with strong performance in real operational environments.

## 3 Dataset Details

As shown in Figure 1,**Aeolus** is a multi-modal dataset designed to address the limitations of traditional flight delay datasets. It integrates three key modalities: tabular data, temporal data in the form of flight chains, and graph-based data representing flight networks. The dataset contains features such as flight times, airport codes, and historical delay information in tabular format, as well as temporal sequences of flights (flight chains) that capture the dynamic nature of delays. Additionally, the dataset includes a flight network structure that models the interconnections between flights and airports, which is crucial for understanding how delays propagate through the system.

Table 2: Comparison of Current Tabular Datasets.

| Benchmark | Dataset Sizes | | Issues (Issues / Datasets) | | | Time-split | Flight-Delay |
|---|---|---|---|---|---|---|---|
| | Samples | Features | Data-Leak | Synthetic | Non-Tab | | |
| [13] | 16,679 | 13 | ✓ | ✓ | ✓ | ✗ | ✗ |
| Tabrizia([24]) | 3,087 | 23 | ✓ | ✓ | ✓ | ✗ | ✗ |
| WildTab ([19]) | 54,943 | 10 | ✓ | ✓ | ✗ | ✗ | ✗ |
| TableShift ([10]) | 840,582 | 23 | ✗ | ✗ | ✗ | ✗ | ✗ |
| [12] | 57.909 | 20 | ✓ | ✓ | ✗ | ✗ | ✗ |
| TabRed [30] | 7,163,150 | 261 | ✗ | ✗ | ✗ | ✓ | ✗ |
| Aeolus (ours) | 54,674,003 | 22 | ✗ | ✗ | ✗ | ✓ | ✓ |

This section focuses on the core components of dataset construction. For implementation details including feature engineering, temporal sequence alignment, and graph merging protocols, please refer to Appendix A.

## 3.1 Tabular Data: Features

We obtain flight data from the U.S. Department of Transportation's Bureau of Transportation Statistics (BTS)[2], including flight statuses, delay information, and airport details, and acquire meteorological data from Meteostat[3]. Specifically, we integrate flight data, airport data, and weather data to create a feature-rich flight delay dataset. The flight and airport data are sourced from the BTS official portal, spanning from 2016 to 2024, with raw data containing 56,668,600 flight records. These records include fields such as date, airline carrier, flight number, route, flight schedules, and delay status. Airport data comprises airport codes, full names, cities, states, countries, latitudes, and longitudes. Weather data obtained from Meteostat includes hourly measurements of temperature, precipitation, wind speed, and atmospheric pressure. Through data matching, temporal alignment, and processing of anomalous data and missing values, we ultimately construct a multi-source integrated flight delay dataset.

## 3.2 Temporal Data: Flight Chains

Flight chains refer to dynamic sequences formed when the same aircraft consecutively executes multiple flight missions. Their core value lies in capturing temporal propagation patterns of delays within the same aircraft's operational sequence. For instance, if the delay duration of a preceding flight exceeds the minimum turnaround time, it inevitably causes delays in subsequent flights.

The structure of a flight chain is illustrated in Figure 2. Each node in the chain represents a flight. Taking a 24-hour operational sequence like "Dallas-Chicago-Atlanta-Los Angeles-New York" as an example, we define the first departure airport within the time window (24 hours in this study) as the primary airport, and the corresponding initial flight (e.g., "Dallas-Chicago") as the primary flight. Subsequent airports and flights are hierarchically labeled as secondary, tertiary, etc.

To construct flight chain datasets, we adopt daily segmentation due to the limited quantity and lower delay rates of overnight cross-day flights. Specifically, for efficient chain generation, we pre-sort flight data by quadruple (OP_CARRIER, OP_CARRIER_FL_NUM, DATE, CRS_DEP_TIME) in ascending order to ensure spatiotemporal continuity. Dynamic grouping is then performed using (OP_CARRIER, OP_CARRIER_FL_NUM, DATE) as identifiers, guaranteeing each chain corresponds to a single aircraft's 24-hour mission sequence. Spatial continuity filters are applied to group entries, where adjacent flights must satisfy destination-to-origin airport consistency (e.g., chain A→B→C→D requires B's origin = A's destination). Incompatible entries form separate chains. To

---

[2]https://www.bts.gov/
[3]https://meteostat.net/en/

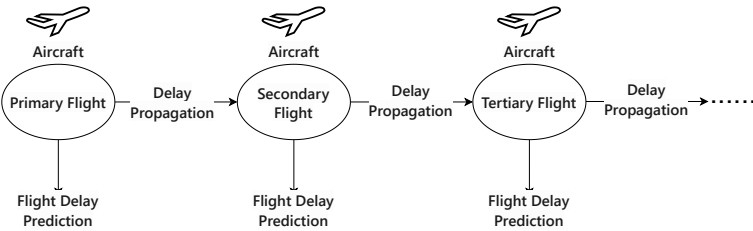

Figure 2: Structure of Flight Chain.

align with deep learning models, sequence length standardization is implemented by setting maximum sequence length $L_{max}$, with truncation/padding applied to overlong/short chains.

### 3.3 Graph Data: Flight Networks

While flight chains effectively analyze temporal delay propagation within individual aircraft sequences, they neglect cross-aircraft delay transmissions caused by shared airport resources, crew scheduling, or passenger transfers.

The construction of flight networks can be considered as a gradual extension from flight chains. Within a specific time window, starting from an individual aircraft, we augment flight chains by adding new edges to represent spatiotemporal delay propagation relationships between different aircraft caused by adjacent temporal-spatial occupancy at the same airport. This transforms the chain structure into a tree structure, forming a flight tree originating from a single aircraft. As shown in Figure 3a, the flight tree constructed is illustrated. For clarity, we employ Tier-1, Tier-2 descriptors to represent Primary Flight, Secondary Flight, etc. The tree contains distinct edge types: red edges indicate delay propagation within the same aircraft, while black edges represent cross-aircraft delay propagation.

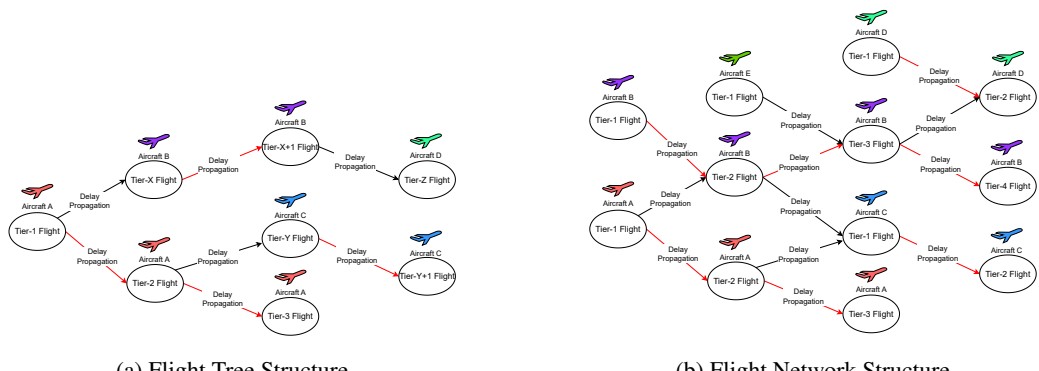

(a) Flight Tree Structure                    (b) Flight Network Structure

Figure 3: Structure of Flight Trees Constructed from Different Root Flights.

Aggregating flight trees within specific spatiotemporal ranges forms flight networks, as shown in Figure 3b. Depending on selected time windows and flight ranges, the network may contain one or multiple weakly connected components. The mathematical formulation of the merging process is:

$$G = \bigcup_{i \in \mathcal{A}} Tree_i = \left( \bigcup_{i \in \mathcal{A}} V(Tree_i), \bigcup_{i \in \mathcal{A}} E(Tree_i) \right) \tag{1}$$

where $\mathcal{A}$ denotes the set of all target aircraft, $V(Tree_i)$ represents the node set of flight tree $Tree_i$ rooted at aircraft $i$, and $E(Tree_i)$ contains two edge types: intra-aircraft delay propagation edges and cross-aircraft delay propagation edges. The resulting network $G$ has a node set as the union of all $V(Tree_i)$ and an edge set as the union of all $E(Tree_i)$.

Algorithmically, each flight serves as a root node for depth-first search to generate its flight tree. Daily aggregation of all trees forms the flight network. A visit tracking array is implemented during traversal to avoid redundant node processing and reduce time complexity.

## 4  Use Cases

To fully exploit the rich information and diversity of the Aeolus dataset, we designed three experimental scenarios: benchmarking tabular models, evaluating time series models, and embedding graph-based models. Each scenario targets a different facet of flight delay prediction, spanning from traditional statistical approaches to advanced deep learning methods.

### 4.1  Benchmarking Tabular Feature Modeling

**Problem Definition.** Accurately predicting flight arrival and departure delays is critical for both airlines and passengers. In this study, we focus solely on tabular features from the Aeolus dataset. These features encompass not only basic flight information but also influential factors such as weather conditions and airport status — all represented in structured, tabular form.

**Setup.** We selected 287,845 flight samples between June 1 and June 15, 2024, a period of high air traffic that can provide rich delay-related information. The dataset contains 8 categorical features and 14 continuous features, a total of 22 features, covering flight information, weather, and other delay-related factors. We employed a temporally-stratified 6:2:2 partitioning strategy to divide the data into training, validation, and test sets, maintaining chronological continuity to prevent data leakage. For detailed analysis of temporal distribution shifts and their impact on model generalization, see Appendix C.

We define two prediction targets: **ARR_Delay** and **DEP_Delay**, and design three tasks for each target with corresponding evaluation metrics. For benchmarking, we select three representative baseline tabular models — **MLP** [31], **ResNet** [11], and **AutoInt** [38] — along with five recent and high-performing tabular models: **FTTransformer** [11], **Tangos** [16], **TabulaRNN** [35], and **SAINT** [33]. We appropriately adapt the architectures and hyperparameters of each method to optimize their performance. Detailed experimental settings can be found in Appendix B.

Table 3: Model Performance Comparison (ARR & DEP). The best results are highlighted in green, second-best in yellow, and worst in red.

| Model | Regressor | | | | Classifier | | | | Distribution | |
|---|---|---|---|---|---|---|---|---|---|---|
| | MSE | | MAE | | AUC | | ACC | | CRPS | |
| | ARR | DEP | ARR | DEP | ARR | DEP | ARR | DEP | ARR | DEP |
| MLP | 0.937 | 0.940 | 0.634 | 0.608 | 0.600 | 0.627 | 0.689 | 0.743 | 0.658 | 0.650 |
| AutoInt | 0.942 | 0.964 | 0.669 | 0.556 | 0.623 | 0.617 | 0.708 | 0.718 | 0.667 | 0.693 |
| ResNet | 0.949 | 1.005 | 0.657 | 0.627 | 0.557 | 0.569 | 0.750 | 0.760 | 0.672 | 0.633 |
| FTTransformer | 0.914 | 0.946 | 0.618 | 0.590 | 0.562 | 0.577 | 0.763 | 0.759 | 0.702 | 0.680 |
| Tangos | 1.038 | 0.992 | 0.690 | 0.613 | 0.616 | 0.627 | 0.679 | 0.759 | 0.655 | 0.657 |
| TabulaRNN | 0.925 | 0.942 | 0.631 | 0.588 | 0.559 | 0.627 | 0.772 | 0.761 | 0.637 | 0.673 |
| SAINT | 0.981 | 0.989 | 0.635 | 0.618 | 0.562 | 0.573 | 0.772 | 0.759 | 0.695 | 0.711 |

**Result Analysis.** As shown in Table 3, the experimental results reveal distinct performance patterns across tasks and delay types. FTTransformer excels in arrival delay regression (MSE: 0.914, MAE: 0.618), demonstrating superior feature interaction modeling, while TabulaRNN achieves top classification accuracy on both delay types (ACC: 0.772/0.761). Significant performance variations between arrival and departure predictions highlight domain adaptation challenges, with MLP showing strong departure regression (MSE: 0.940) but mediocre arrival performance.

Notably, while models attain high accuracy (0.74-0.77), their modest AUC scores (<0.63) indicate persistent class imbalance issues. TabulaRNN emerges as the most robust model with consistent top-tier performance, whereas SAINT struggles with uncertainty quantification (highest CRPS). These

findings suggest that optimal delay prediction requires task-specific model selection, potentially combining FTTransformer's regression strengths with TabulaRNN's classification capabilities. The results underscore the need for specialized approaches addressing domain shift and imbalance challenges in aviation prediction systems.

## 4.2 Benchmarking Sequential Modeling

**Problem Definition.** In this section, we perform classification experiments using arrival delays (ARR_Delay) and departure delays (DEP_Delay). Given the flight chain structure $\mathcal{C} = \{f_1, f_2, \ldots, f_T\}$, where each node $f_t$ represents the $t$-th flight in the chain, the corresponding static features are denoted as $\mathbf{x}_t \in \mathbb{R}^{m+n}$. Here $\mathbf{x}_t = \{c_1, \ldots, c_m, z_1, \ldots, z_n\}$ consists of $m$ categorical features and $n$ numerical features. The research objective of this paper is to establish a mapping function $F : \{\mathbf{x}_t\}_{t=1}^T \to \{\hat{y}_t\}_{t=1}^T$ that predicts the delay probability $\hat{y}_t \in [0, 1]$ for each flight in the chain based on its static feature inputs.

**Setup.** We utilized the full-year 2024 data containing 6,284,841 raw records. In this section, we employed 8 categorical features and 7 numerical features, totaling 15 features, covering flight information, weather, and other delay-related factors. The dataset is divided by days with a ratio of 6:2:2, with the guarantee that every month contains at least one day allocated to the training, validation, and test sets respectively. The final partition comprises 193 days for training, 72 days for validation, and 70 days for testing.

In the experiment, we use **label encoding** for categorical features and **normalization** for numerical features. We selected two types of basic sequential models (**LSTM[15]**, **GRU [6]**) and two types of enhanced hybrid architectures (**CNN-LSTM[9]**, **MogrifierLSTM[25]**) for experiments. For different methods, we made appropriate structural and hyperparameter adjustments to achieve the best results. For more details about the experimental settings, please refer to Appendix B.

**Result Analysis.** The stable AUC performance across all models (ARR: 68.5-69.0%, DEP: 68.8-69.5%) validates the effectiveness of flight chain construction in preserving temporal dependencies, with <1.5% variance indicating robust capture of essential propagation patterns. Departure delays consistently achieve higher AUC than arrival delays (max 69.54% vs 68.99%), reflecting stronger systematic propagation through maintainable factors versus unpredictable en-route impacts. The minimal performance variance across architectures (<0.3% AUC gap) further demonstrates flight chain structure dominance over model selection, with MogrifierLSTM showing marginal advantage in departure delay prediction.

Table 4: Model Comparison Results of ARR_Delay and DEP_Delay in Flight Chain Scenarios.

| Model | ARR.Delay | | | | | DEP.Delay | | | | |
|---|---|---|---|---|---|---|---|---|---|---|
| | Accuracy | Recall | Precision | F1 | AUC | Accuracy | Recall | Precision | F1 | AUC |
| LSTM | 0.6341 | 0.2932 | 0.6434 | 0.4029 | 0.6899 | 0.6335 | 0.2925 | 0.6487 | 0.4032 | 0.6928 |
| GRU | 0.6342 | 0.2924 | 0.6389 | 0.4012 | 0.6876 | 0.6316 | 0.2929 | 0.6581 | 0.4054 | 0.6951 |
| CNN-LSTM | 0.6395 | 0.2941 | 0.6280 | 0.4006 | 0.6851 | 0.6305 | 0.2901 | 0.6466 | 0.4005 | 0.6882 |
| MogrifierLSTM | 0.6305 | 0.2912 | 0.6460 | 0.4014 | 0.6873 | 0.6419 | 0.2969 | 0.6407 | 0.4058 | 0.6954 |

## 4.3 Benchmarking Graph Modeling

**Problem Definition.** In this section, we perform classification experiments using arrival delays (ARR_Delay). The flight network is represented as a graph $\mathcal{G} = (V, E)$, where nodes $v_i \in V$ correspond to individual flights, and directed edges $e_{ij} \in E$ are formed between two flights when spatiotemporal correlations exist, indicating potential delay propagation relationships. In this section, we evaluate whether the features extracted by the GNN contribute to improving the accuracy of flight delay prediction.

**Setup.** Following the experimental protocol established in Section 4.2 regarding data partitioning (6:2:2 ratio), feature preprocessing (15 total features), and hyperparameter tuning strategies, we focus here on graph-specific implementations. We employ node embedding method **VGAE[18]** to extract topological attributes and spatiotemporal correlations of flight nodes through the flight

network structure, integrating these embeddings with static variables via **AFM[41]**. Comparisons are made against baseline **AFM** and **AFM+DeepWalk[27]** embedding approaches.

**Result Analysis.** The integration of flight network embeddings significantly enhances prediction performance, with AFM+VGAE achieving the highest AUC (67.94%) compared to baseline AFM (67.23%) and AFM+DeepWalk (67.65%). Both embedding methods demonstrate effectiveness - DeepWalk improves AUC by 0.42% through airport proximity modeling, while VGAE's 0.71% total gain confirms its superior capability in capturing multi-hop delay propagation via structural learning. The 0.29% AUC difference between VGAE and DeepWalk highlights the advantage of probabilistic graph embeddings over random walk-based approaches for modeling complex delay cascades.

Table 5: Comparison of Node Embedding Methods with AFM Framework.

| Model | Classification Metrics | | | | |
|---|---|---|---|---|---|
| | Accuracy | Recall | Precision | F1 | AUC |
| AFM (Baseline) | 0.6441 | 0.2602 | 0.5972 | 0.3624 | 0.6723 |
| AFM + DeepWalk | 0.6487 | 0.2638 | 0.6009 | 0.3672 | 0.6765 |
| AFM + VGAE | 0.6526 | 0.2675 | 0.6023 | 0.3705 | 0.6794 |

# 5 Temporal Distribution Shift Analysis

Building upon our discussion of temporal splitting strategies in 2.3, we conduct an in-depth analysis of temporal distribution shifts in flight delay data. This investigation addresses two critical aspects: (1) the methodological implications of split strategies on model evaluation, and (2) the substantive impact of exogenous shocks on delay patterns.

## 5.1 Methodological Implications of Split Strategies

Our temporal splitting approach, as detailed in 4.1, represents a deliberate departure from conventional random splitting methodologies. To quantitatively demonstrate the necessity of this approach, we present a comprehensive ablation study comparing the two partitioning strategies across seven state-of-the-art tabular models.

Table 6: Comparative Analysis of Temporal versus Random Splitting Strategies on ARR_DELAY Prediction

| Model | AUC (Temporal) | ACC (Temporal) | AUC (Random) | ACC (Random) |
|---|---|---|---|---|
| MLP | 0.600 | 0.689 | 0.645 | 0.688 |
| AutoInt | 0.623 | 0.708 | 0.671 | 0.724 |
| ResNet | 0.557 | 0.750 | 0.640 | 0.782 |
| FTTransformer | 0.562 | 0.763 | 0.623 | 0.758 |
| Tangos | 0.616 | 0.679 | 0.639 | 0.783 |
| TabulaRNN | 0.559 | 0.772 | 0.653 | 0.798 |
| SAINT | 0.562 | 0.772 | 0.600 | 0.728 |
| **Average** | 0.582 | 0.733 | 0.639 | 0.751 |

The results in Table 6 reveal a systematic performance inflation under random splitting conditions, with an average AUC increase of 0.057 across all models. This phenomenon underscores the presence of significant temporal leakage when future information is inadvertently incorporated during training. The consistency of this pattern across diverse architectural paradigms—from traditional MLPs to advanced transformer-based models—suggests that temporal distribution shifts represent a fundamental characteristic of flight delay data rather than a model-specific artifact.

## 5.2 Exogenous Shock Analysis: COVID-19 Impact

Beyond methodological considerations, we investigate the substantive impact of exogenous shocks on temporal distribution patterns. The COVID-19 pandemic provides a natural experiment for examining how large-scale disruptions affect delay propagation dynamics and model generalization.

We analyze three distinct pandemic phases using carefully constructed temporal splits:

- **Severe disruption phase** (March-August 2020): Characterized by unprecedented reductions in air traffic and atypical delay patterns
- **Moderate disruption phase** (July-August 2020): Representing partial recovery with residual operational anomalies
- **Post-disruption validation** (September-October 2020): Serving as a consistent evaluation benchmark

Table 7: Quantifying COVID-19 Induced Distribution Shifts on Model Performance

| Training Period | Model | ARR AUC | ARR ACC |
|---|---|---|---|
| **Severe Disruption** | CatBoost | 0.6057 | 0.6065 |
| (Mar-Aug 2020) | MLP | 0.5328 | 0.5446 |
| | AutoInt | 0.5308 | 0.5589 |
| **Moderate Disruption** | CatBoost | 0.6158 | 0.6319 |
| (Jul-Aug 2020) | MLP | 0.5528 | 0.6536 |
| | AutoInt | 0.5540 | 0.6179 |
| **Performance** | | **+0.023** | **+0.038** |

As demonstrated in Table 7, models trained exclusively on severe disruption periods exhibit markedly inferior performance compared to those trained on moderate disruption data, with an average performance improvement of 0.023 AUC and 0.038 ACC when excluding the most anomalous months. This degradation pattern persists across diverse algorithmic approaches, indicating that temporal distribution shifts induced by exogenous shocks fundamentally alter the underlying data generating process.

## 6 Conclusion

In this paper, we introduced **Aeolus**, a unified benchmark for flight delay prediction that addresses the disconnect between academic tabular learning research and real-world deployment challenges. Unlike existing datasets that rely on flat, simplified feature sets, Aeolus incorporates rich, multimodal data—including tabular attributes, temporal aircraft rotations, and dynamic graph-structured airport interactions—to more faithfully reflect the operational complexity of air transportation systems. We hope Aeolus will foster more realistic, multimodal, and generalizable approaches to tabular machine learning. Future work may extend the benchmark to incorporate additional signals (e.g., crew rosters, passenger itineraries), enrich graph dynamics, and support cross-domain generalization studies to further advance delay forecasting and industrial ML research.

## Acknowledgments and Disclosure of Funding

We would like to thank the support from the National Natural Science Foundation of China under Grants No. 62406206, 62402414, 62273244, and 72201184. We also extend our sincere appreciation to the providers of the datasets used in this study, whose contributions were essential to the experimental validation of our proposed methods. Furthermore, we thank the anonymous reviewers for their insightful comments and suggestions, which significantly improved the quality of this manuscript. Special thanks are due to our colleagues and collaborators for their valuable discussions and technical support throughout this research endeavor.

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

# A    More Dataset Details

We provide a comprehensive supplementary introduction to the **Aeolus** dataset in this section, covering data composition, statistics, visualization and analysis.Lastly, We conclude by presenting statements on data availability and access links.

## A.1    Data Composition and Statistics

Table 8 presents the yearly statistics of the Aeolus dataset from 2016 to 2024, detailing the number of flight samples along with the mean and standard deviation of three critical features: arrival delay (ARR_DELAY), departure delay (DEP_DELAY), and air time (AIR_TIME). This table demonstrates the dataset's extensive temporal coverage and reveals important operational variations over time, such as the marked reduction in average delays during the COVID-19 pandemic in 2020 and the subsequent recovery to typical delay levels in the following years. Meanwhile, the relative stability of air time across years indicates consistent scheduled flight durations throughout the dataset.

Building on this, Table 9 provides a systematic classification of the dataset's features into categorical and continuous types. The categorical features consist of identifiers and temporal attributes including airline carrier codes, flight numbers, date components, and airport codes. In contrast, continuous features cover scheduled departure and arrival times, scheduled flight durations, various weather measurements at both origin and destination airports, as well as geographic coordinates. This comprehensive feature set integrates operational and environmental information, enabling rich modeling of the multifaceted factors influencing flight delays.

Together, Table 8 and Table 9 offer a thorough overview of Aeolus's data composition and statistical characteristics. The large-scale, multimodal nature of the dataset, combined with detailed feature categorization and realistic temporal variations, establishes Aeolus as a valuable benchmark for flight delay prediction research. This foundation supports the development of robust models capable of capturing the complex, dynamic, and context-dependent phenomena inherent in real-world air traffic systems.

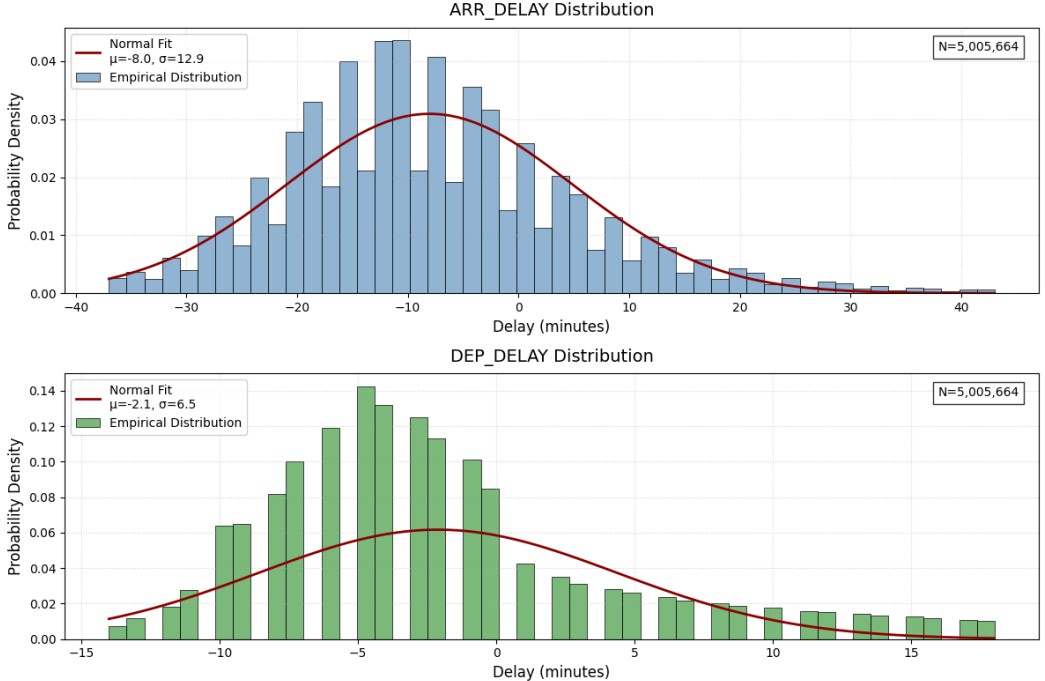

Figure 4: Histogram and normal fit of ARR_DELAY and DEP_DELAY in 2024.

Table 8: Aviation Data Statistics (2016-2024).

| Year | Samples | ARR_DELAY | | DEP_DELAY | | AIR_TIME | |
|---|---|---|---|---|---|---|---|
| | | Mean | Std | Mean | Std | Mean | Std |
| 2016 | 5,537,987 | 3.519 | 41.873 | 8.874 | 39.599 | 116.526 | 73.529 |
| 2017 | 5,575,872 | 4.322 | 45.854 | 9.658 | 43.632 | 117.521 | 74.279 |
| 2018 | 6,986,842 | 4.998 | 46.781 | 9.867 | 44.476 | 111.699 | 71.271 |
| 2019 | 7,161,827 | 5.379 | 50.875 | 10.847 | 48.581 | 111.816 | 70.855 |
| 2020 | 4,312,091 | −5.108 | 37.686 | 1.970 | 35.429 | 109.326 | 67.005 |
| 2021 | 5,755,666 | 2.949 | 48.928 | 9.276 | 46.997 | 113.649 | 69.823 |
| 2022 | 6,413,416 | 6.934 | 54.136 | 12.503 | 52.119 | 113.146 | 70.605 |
| 2023 | 6,645,461 | 6.569 | 56.709 | 12.186 | 54.703 | 115.001 | 70.998 |
| 2024 | 6,284,841 | 7.108 | 57.675 | 12.590 | 55.510 | 114.982 | 70.588 |

Table 9: Flight Data Feature Classification.

| Category | Feature | Description |
|---|---|---|
| Categorical | OP_CARRIER | Airline carrier code (e.g., AA for American Airlines) |
| | OP_CARRIER_FL_NUM | Unique flight number assigned by the carrier |
| | FL_YEAR | Year when the flight occurred (e.g., 2024) |
| | FL_MONTH | Month of the flight (1-12) |
| | FL_DAY | Day of the month (1-31) |
| | FL_WEEK | Week of the year (1-52) |
| | ORIGIN_INDEX | Unique code identifying departure airport |
| | DEST_INDEX | Unique code identifying destination airport |
| Continuous | CRS_DEP_TIME_MIN | Scheduled departure time in minutes since midnight |
| | CRS_ARR_TIME_MIN | Scheduled arrival time in minutes since midnight |
| | CRS_ELAPSED_TIME | Scheduled flight duration in minutes |
| | FLIGHTS | Number of flights (typically 1) |
| | O_TEMP | Temperature at departure airport in Fahrenheit |
| | O_PRCP | Precipitation at departure airport in inches |
| | O_WSPD | Wind speed at departure airport in mph |
| | D_TEMP | Temperature at destination airport in Fahrenheit |
| | D_PRCP | Precipitation at destination airport in inches |
| | D_WSPD | Wind speed at destination airport in mph |
| | O_LATITUDE | Latitude coordinate of departure airport |
| | O_LONGITUDE | Longitude coordinate of departure airport |
| | D_LATITUDE | Latitude coordinate of destination airport |
| | D_LONGITUDE | Longitude coordinate of destination airport |
| Target | DEP_DELAY | Actual departure delay in minutes (positive for delays, negative for early departures) |
| | ARR_DELAY | Actual arrival delay in minutes (positive for delays, negative for early arrivals) |

## A.2 Data Visualization and Analysis

Figure 4 illustrates the statistical distributions of ARR_DELAY and DEP_DELAY in the Aeolus 2024 dataset. Both histograms exhibit heavy-tailed and right-skewed characteristics, highlighting the prevalence of minor delays with occasional severe outliers. Normal distribution curves are superimposed for comparison, revealing significant deviation from Gaussian assumptions. These findings underscore the non-symmetric nature of the delay data and motivate the use of robust loss functions and probabilistic evaluation metrics, such as CRPS, when modeling delay durations. Additional visualizations, including delay propagation patterns and SHAP analysis, are available in the project repository at the repository `https://github.com/Flnny/Delay-data`

## A.3  Data Statements and Accountability

The code and data used for the experiment can be accessed in the repository `https://github.com/Flnny/Delay-data`. The official dataset will be hosted on the Kaggle repository `https://www.kaggle.com/datasets/flnny123/mfddmulti-modal-flight-delay-dataset/data`. Our code and dataset follow the CC BY-NC 4.0 International License. All authors confirm the data license and commit that the dataset will only be used for academic research.

# B  More Experimental Details

## B.1  Hardware configuration and Training time

Our running environment consists of a Windows server equipped with 1×NVIDIA RTX 4070(12GB)GPU.To carry out benchmark testing experiments, all baselines are set to run for a duration of 24 hours by default, with specific timings contingent upon the method.

## B.2  Flight Table Experiment

**Detail Datasets.** As shown in Table 9, we provide the statistical information of the datasets used in this experiment. To further illustrate, we give an intuitive visualization as shown in Figure 4.

**Detail Baselines.** We set two prediction targets: **ARR_Delay** and **DEP_Delay**, and designed three tasks for each target:

- **Regression task**: Predict the specific delay time, using **Mean Squared Error (MSE)** and **Mean Absolute Error (MAE)** as evaluation indicators.

- **Classification task**: Determine whether the flight is delayed (more than 15 minutes is considered a delay), using **Area Under the Curve (AUC)** and **Accuracy (ACC)** as evaluation indicators.

- **Uncertainty prediction task**: Based on the regression task setting, the **Continuous Ranked Probability Score (CRPS)** is used as the evaluation indicator.

In the experiment, we use **label encoding** for discrete features and **normalization** for continuous features. To adapt to different task requirements, we **standardized the target delay time** in the regression task. We selected two types of basic table models (**MLP**, **ResNet**) and five types of the latest and most effective table models (**AutoInt**, **FTTransformer**, **Tangos**, **TabulaRNN**, **SAINT**) for experiments.Below is a brief introduction to each method:

- **AutoInt [38]**It automatically learns high-order interactions between features through the self-attention mechanism, which is particularly suitable for processing high-dimensional and sparse tabular data; its multi-head attention structure can also provide a certain degree of model interpretability.

- **FTtransformer [11]**The Transformer architecture is successfully applied to tabular data modeling, effectively capturing complex dependencies between features through feature tokenization and layer normalization; experiments show that it outperforms the gradient boosting tree model on most tabular tasks

- **MLP [31]**As the most basic feedforward neural network, it realizes nonlinear mapping by stacking fully connected layers. Although it has a simple structure, it can still handle a variety of classification and regression tasks with the help of modern optimizers.

- **Tangos [16]**A regularization method designed specifically for tabular data that improves model generalization through feature masking and contrastive learning; its core idea is to constrain the sensitivity of the neural network to changes in irrelevant features.

- **TabulaRNN [35]**Innovatively introduces RNN sequence modeling capabilities into tabular data processing, capturing dynamic relationships between features through time step expansion; combined with the attention mechanism, the time evolution pattern of key features can be identified

- **SAINT [33]** A phased self-attention mechanism is used to process tabular data, first learning feature embedding independently and then modeling interaction relationships; this separation structure significantly improves the stability of model training.
- **ResNet [11]** An adaptation of the ResNet architecture, suitable for tabular data applications.

We have chosen Mambular + https://github.com/basf/mamba-tabular, a benchmark toolbox designed for tabular deep learning model prediction, as our code framework. For all methods, we followed the original default parameter settings, set max epoch=50, learning rate=0.001, adopted early stopping strategy and set patience=5, and made appropriate adjustments to obtain the best performance.

**Detail Metrics.** Our assessment is performed on renormalized datasets, with regression tasks evaluated using metrics including Mean Absolute Error (MAE) and Mean Squared Error (MSE), classification tasks assessed via Area Under the Curve (AUC) and Accuracy (ACC), and distributional tasks quantified through the Continuous Ranked Probability Score (CRPS). Formally, assuming $n$ represents the number of observed samples, $y_i$ denotes the $i$-th actual sample, and $\hat{y}_i$ is the corresponding prediction, these metrics are formulated as follows:

- **Mean Absolute Error (MAE):**

$$MAE = \frac{1}{n} \sum_{i=1}^{n} |y_i - \hat{y}_i|$$

  where $| \cdot |$ denotes absolute value. MAE measures the average magnitude of errors between predictions and observations, treating all deviations equally.

- **Mean Squared Error (MSE):**

$$MSE = \frac{1}{n} \sum_{i=1}^{n} (y_i - \hat{y}_i)^2$$

  MSE computes the average squared prediction errors, emphasizing larger errors through squaring. Its square root (RMSE) preserves units.

- **Area Under ROC Curve (AUC):**

$$AUC = \int_0^1 TPR(FPR^{-1}(r)) \, dr$$

  where $TPR$ (True Positive Rate) and $FPR$ (False Positive Rate) are functions of the classification threshold. AUC evaluates binary classifier performance across all possible thresholds (1.0 = perfect, 0.5 = random).

- **Accuracy (ACC):**

$$ACC = \frac{TP + TN}{TP + TN + FP + FN}$$

  with $TP$ (True Positives), $TN$ (True Negatives), $FP$ (False Positives), and $FN$ (False Negatives). ACC measures the proportion of correct classifications among all cases.

- **Continuous Ranked Probability Score (CRPS):**

$$CRPS = \int_{-\infty}^{\infty} \left( F(y) - \mathbb{1}\{y \geq y_{\text{obs}}\} \right)^2 dy$$

  where $F(y)$ is the predicted CDF and $\mathbb{1}$ is the indicator function. CRPS quantifies the difference between predicted and empirical cumulative distributions for probabilistic forecasts.

## B.3 Flight Chain Experiment

**Detail Baselines.** We selected two types of basic sequential models (**LSTM**, **GRU**) and two types of enhanced hybrid architectures (**CNN-LSTM**, **MogrifierLSTM**) for experiments. Below is a brief introduction to each method:

- **LSTM [15]** A recurrent neural network architecture with memory cells and gating mechanisms to capture long-term dependencies in sequential data. The forget gate structure helps mitigate gradient vanishing issues in flight chain modeling.

- **GRU [6]** A simplified variant of LSTM that combines hidden state and cell state through update and reset gates. Suitable for modeling short-term temporal patterns in flight delay propagation.
- **CNN-LSTM [9]** Hybrid architecture combining convolutional layers for local pattern extraction and LSTM for temporal dynamics modeling. The CNN component captures spatial correlations within flight sequences.
- **MogrifierLSTM [25]** Enhanced LSTM with iterative interactions between input and hidden states through alternating linear transformations. Improves feature modulation for flight sequence analysis.

For all methods, we followed the original default parameter settings, set max epoch=50, learning rate=0.001, adopted early stopping strategy and set patience=5, and made appropriate adjustments to obtain the best performance.

**Detail Metrics.** Our assessment focuses on binary classification of flight delays (threshold: 15 minutes), evaluated through five metrics: Accuracy (ACC), Recall (True Positive Rate), Precision (Positive Predictive Value), F1 Score, and Area Under the ROC Curve (AUC). Formally, assuming $n$ represents the number of observed samples, these metrics are formulated as follows:

- **Accuracy (ACC):**
$$ACC = \frac{TP + TN}{TP + TN + FP + FN}$$

    where $TP$ (True Positives), $TN$ (True Negatives), $FP$ (False Positives), and $FN$ (False Negatives) are derived from the confusion matrix. ACC measures the overall prediction correctness.

- **Recall (TPR):**
$$TPR = \frac{TP}{TP + FN}$$

    Quantifies the proportion of actual delayed flights correctly identified. Critical for air traffic control resource planning.

- **Precision (PPV):**
$$PPV = \frac{TP}{TP + FP}$$

    Evaluates the reliability of positive predictions to minimize false alarms in airport operations.

- **F1 Score:**
$$F1 = \frac{2 \cdot PPV \cdot TPR}{PPV + TPR}$$

    Harmonic mean balancing precision and recall trade-offs for imbalanced delay classification.

- **Area Under ROC Curve (AUC):**
$$AUC = \int_0^1 TPR(FPR^{-1}(r)) \, dr$$

    where $TPR$ (True Positive Rate) and $FPR$ (False Positive Rate) are functions of the classification threshold. AUC evaluates ranking performance across all thresholds (1.0 = perfect, 0.5 = random).

## B.4 Flight Network Experiment

**Detail Baselines.** We implement a two-stage training paradigm: (1) Generating flight node embeddings from the flight network topology, (2) Integrating embeddings with static features for tabular prediction. The core components include:

- **AFM [41]** A hybrid prediction architecture combining factorization machines with attention mechanisms. Learns feature interactions through adaptive weight allocation, particularly effective for sparse feature scenarios in delay prediction.

- **Deep Walk [27]** Unsupervised graph embedding method based on random walks. Preserves node proximity by simulating truncated walks on the flight network to generate topological representations.
- **VGAE [18]** Graph neural architecture combining encoder-decoder framework with variational inference. Learns probabilistic embeddings by modeling structural connectivity in the flight network.

For all methods, we followed the original default parameter settings, set max epoch=100, learning rate=0.001, adopted early stopping strategy with patience=5, and made appropriate adjustments to obtain the best performance. The implementation pipeline comprises:

1. *Embedding Generation Phase*: Train graph embedding methods on flight network topology
2. *Prediction Phase*: Combine node embeddings with static features as AFM input

**Detail Metrics.** Consistent with Section B.3, we evaluate binary delay classification (threshold: 15 minutes) using Accuracy (ACC), Recall (TPR), Precision (PPV), F1 Score, and AUC metrics. Formal definitions remain identical to those specified in Section B.3.

# C   More Discussion

While Aeolus represents a significant advancement in multimodal flight delay benchmarking, we identify several limitations that warrant discussion and future research directions.

**Temporal Coverage Bias.** The dataset's 9-year timeframe (2016-2024) includes the anomalous COVID-19 pandemic period (2020-2021), where global air traffic dropped by 53.3% (ICAO 2021). This introduces non-stationarity in delay patterns—average departure delays decreased from 10.85 minutes (2019) to 1.97 minutes (2020) as shown in Table 5. Though we include temporal splits to mitigate distribution shift, models trained on this period may learn atypical operational patterns. Future versions could benefit from: (1) Pandemic-specific evaluation subsets, (2) Counterfactual analysis of pre/post-COVID delay propagation mechanisms.

**Geographic Representation.** Despite covering 320 airports (Table 1), Aeolus exhibits three geographic biases: (1) 78.4% of flights originate from North America due to BTS data sourcing, (2) Major hubs (e.g., ATL, ORD) are overrepresented (top 20 airports account for 41.7% of flights), and (3) Developing regions (Africa, South Asia) are underrepresented. This limits generalizability to global operations where airport infrastructure heterogeneity affects delay dynamics. A promising extension would be integrating EUROCONTROL and CAAC data for multinational coverage.

**Feature Granularity Constraints.** Current operational features lack three critical dimensions: (1) Real-time air traffic control (ATC) decisions (e.g., ground stops, flow restrictions), (2) Aircraft-specific maintenance histories, and (3) Passenger flow data. These omissions restrict models from capturing micro-level delay catalysts. Our flight network edges approximate resource contention but cannot model dynamic gate reassignments or crew scheduling adjustments. Future collaborations with airlines could enable richer feature engineering while addressing privacy concerns through differential privacy techniques.

# D   Broader Impact

**Economic Impact.** Aeolus enables more accurate flight delay prediction, which can help airlines optimize operations, reduce fuel waste from prolonged taxiing or holding patterns, and minimize compensation costs for delayed passengers. However, if models trained on this dataset are deployed without proper validation, they could lead to suboptimal scheduling decisions, exacerbating delays rather than mitigating them.

**Environmental Impact.** By improving delay forecasting, Aeolus could contribute to reducing unnecessary fuel burn and $CO_2$ emissions caused by inefficient flight operations. However, the computational resources required to train large-scale multimodal models on this dataset may offset some of these environmental benefits unless energy-efficient training methods are adopted.

**Social Impact.** Better delay prediction enhances passenger experience by enabling proactive rebooking and reducing uncertainty. However, biases in the dataset (e.g., underrepresentation of

regional airports) may lead to unequal prediction quality across different traveler demographics, disproportionately affecting passengers relying on smaller airports.

**Research Impact.** Aeolus fills a critical gap in multimodal tabular data research, fostering innovation in flight delay modeling and broader structured-data ML. However, its complexity may raise the barrier to entry for researchers without access to high-performance computing resources, potentially limiting reproducibility and equitable participation in this research acrea.

**Regulatory Impact.** Widespread adoption of models trained on Aeolus could influence aviation policies, such as slot allocation and delay compensation rules. Policymakers must ensure such models are transparent and auditable to prevent misuse by airlines seeking to justify operational shortcomings rather than improving service reliability.

**Ethical Impact.** While the dataset itself is anonymized, improper use of predictive models could lead to privacy concerns—for instance, if sensitive operational patterns are reverse-engineered from delay predictions. Care must be taken to ensure compliance with data protection regulations like GDPR when deploying such systems in practice.

