# OpenReview forum: "Aeolus: A Multi-structural Flight Delay Dataset"
_NeurIPS.cc/2025/Datasets_and_Benchmarks_Track — NeurIPS 2025 Datasets and Benchmarks Track poster_

### Official Review · Reviewer_f5F8 · 2025-06-02

**Rating:** 4
**Confidence:** 2

**Summary:**

The authors introduce a multimodal, temporally-aware, and structurally-rich dataset to improve the realism and effectiveness of machine learning models for flight delay prediction and general tabular learning.

**Additional Feedback:**

N/A

**Dataset Code Accessibility:**

Yes

**Dataset Code Comments:**

Yes, the datasets are accessible.

**Ethical Considerations:**

No, there are no or only very minor ethics concerns

**Final Justification:**

After carefully reviewing the authors' responses and other reviewers' comments, I believe this work has its value in providing a novel dataset for the research community, although the corresponding evaluation framework does not seem to be perfect. For instance, the authors also acknowledge that the "Geographic Bias" exists and could affect their results.

After thoroughly weighing the aforementioned strengths and weaknesses, I have decided to maintain my original rating. If the other reviewers are all satisfied, I would not strongly oppose its acceptance.

**Limitations Weaknesses:**

**1. [Important] Temporal distribution bias:** As mentioned in Appendix A, the inclusion of COVID-era data (2020–2021) may introduce atypical delay patterns that may mislead models without dedicated mitigation strategies.

**2. [Important] Geographical bias:** As mentioned in Appendix C, 78.4% of flights originate from North America; underrepresentation of African, South Asian, or regional airports may limit generalizability.

**3. Interestingly low AUC scores:** Despite large-scale data, the models attain AUCs <0.65 in most cases, indicating limited predictive power. I am unsure if this is because of insufficient hyperparameter tuning or the complexity of delay phenomena. I would suggest the authors provide further discussion on this.

**Strengths Contributions:**

The primary strengths are three-fold:

- **Novel dataset:** Introduction of Aeolus, a flight delay dataset covering 54.67 million flights across 9 years and 320 airports, incorporating tabular, sequential (flight chains), and graph (flight networks) modalities.
- **Unified benchmarking framework:** Unified tasks including regression, classification, and uncertainty prediction with baselines across tabular (e.g., FTTransformer), sequence (e.g., GRU, MogrifierLSTM), and graph (e.g., VGAE) models.
- **Good reproducibility:** The provided codebase shows attempts to mitigate data leakage and benchmark protocols ensuring reproducibility.

---

> ### Author Rebuttal · Authors · 2025-07-26
>
> ### **Q1: Temporal Shift Due to COVID Data**
> **A1:**We appreciate the reviewer’s insightful comment regarding the potential time distribution bias. As we have explored in our study of the "Monthly Average Arrival Delay Trends (2019-2022) with COVID-19 Impact", illustrated in the graph(which is also available in the corresponding GitHub repository), we successfully demonstrate the presence of atypical delay patterns during the COVID-19 era. The months of March to June 2020 show an abnormal trend, with the delay values significantly lower than in other years, and the pattern itself differs notably from the trends observed in previous years.
>
> To address this issue, we are conducting experiments using the Aeolus dataset to better understand and quantify the impact of atypical delay patterns. We selected samples from different periods of the COVID-19 pandemic to assess its effect on model performance:
>
> - **March-June 2020 (severe pandemic period)**: 30,000 samples as the "severe pandemic training set."
> - **July-August 2020 (slightly alleviated pandemic)**: 30,000 samples as the "mild pandemic training set."
> - **September-October 2020**: 40,000 samples, equally split into a validation set and a test set.
>
> To compare the pandemic's impact, we tested three classic algorithms—CatBoost, MLP, and AutoInt—using the following training sets:
> 1. Severe pandemic training set + mild pandemic training set
> 2. Mild pandemic training set
>
> The target variable was **ARR_DELAY**, and the results were as follows:
>
> | Train Period               | Model    | ARR AUC | ARR ACC |
> |-------------------------|----------|---------|---------|
> | **March-August** | CatBoost | 0.6057  | 0.6065  |
> |                               | MLP     | 0.5328  | 0.5446       |
> |                               | AutoInt | 0.5308  | 0.5589       |
> | **July-August**     | CatBoost | 0.6158  | 0.6319  |
> |                              | MLP     | 0.5528  | 0.6536       |
> |                              | AutoInt | 0.5540  | 0.6179       |
>
> From these results, we observe that models perform worse when trained on datasets from periods of severe pandemic disruption. This suggests that the atypical delay patterns caused by the pandemic negatively impact model performance. Moreover, the performance gains observed after excluding these periods from the training set indicate that removing highly irregular data can help mitigate the influence of such anomalies. This finding offers valuable guidance for shaping our future research directions.
>
> In response to temporal distribution shift, we are actively exploring methods to handle these periods separately, such as creating pandemic-specific subsets or employing counterfactual analysis. These strategies will ensure that models do not overfit to these abnormal patterns, allowing for better generalization to post-pandemic scenarios. Our work on this will be showcased in future studies, where we will present the results and insights gained from these experiments.
>
>
>
> ---
>
> ### **Q2: Geographic Bias — Overrepresentation of North America**
> **A2:** We appreciate the reviewer’s insightful comment regarding the geographical bias of the Aeolus dataset. As noted in Appendix C, the dataset indeed reflects a significant concentration of flights originating from North America, with approximately 78.4% of the flights coming from this region. This overrepresentation of major hubs, coupled with the underrepresentation of regions like Africa, South Asia, and smaller airports, may limit the generalizability of the model to global operations.
>
> We acknowledge that this geographical bias could affect the robustness of the model in capturing the full spectrum of delay dynamics. To address this, we are actively working on expanding the dataset by incorporating additional regional data sources, such as EUROCONTROL and CAAC, to improve the dataset's geographical diversity. This will help ensure that our models can better generalize to a wider range of airports and regions in future applications.
>
> ---
>
> ### **Q3: Low AUC Performance — Model Limitation or Task Complexity?**
> **A3:** We appreciate the reviewer’s observation regarding the low AUC scores (<0.65) despite the large-scale data used. We believe these values are primarily a result of the intrinsic complexity and high uncertainty inherent in real-world delay prediction, rather than insufficient hyperparameter tuning. For hyperparameter optimization, we used RandomizedSearchCV, which performs a randomized search over the specified parameter space to select the best combination. We conducted 10 experiments using 3-fold cross-validation and selected appropriate scoring metrics to reflect the objectives of the task at hand. Additionally, all models were trained under a unified configuration with early-stopping to ensure consistency and prevent overfitting.
>
> Flight delays are influenced by numerous unobservable or difficult-to-quantify external factors, such as sudden weather changes, passenger connections, gate conflicts, and ATC interventions. These factors cannot be fully captured by the available modeling inputs, which inherently limits predictive performance.
>
> Furthermore, the extracted tabular benchmark from Aeolus reflects real-world distributions, with a delay:non-delay ratio of approximately 1:3. This includes structural noise, delay propagation effects, and temporal drift, which are characteristic of real-world systems. The "moderate accuracy + high realism" approach was intentionally chosen to benchmark models that exhibit true robustness and interpretability, rather than aiming for perfect accuracy, as this more closely mirrors the challenges faced in operational deployment.

---

> > ### Comment · Reviewer_f5F8 · 2025-08-04
> >
> > I would like to thank the authors for their detailed responses. After carefully considering the points the authors presented and other reviewers' comments, I believe that my initial evaluation remains valid. Therefore, my initial positive score remains.

---

> > > ### Author Response · Authors · 2025-08-04
> > >
> > > Thank you for your follow-up and for taking the time to review our work. We appreciate your thoughtful consideration and are glad that you continue to view our submission positively.

---

### Official Review · Reviewer_4X91 · 2025-06-29

**Rating:** 5
**Confidence:** 5

**Summary:**

This paper presents Aeolus, a large-scale, multi-modal dataset for flight delay prediction, covering over 50 million flights across nine years. Aeolus provides three aligned modalities: (i) tabular data with rich flight, weather, and airport features; (ii) temporal data via flight chains to model delay propagation within aircraft sequences; and (iii) graph data capturing cross-aircraft dependencies via shared airport resources. The benchmark supports regression, classification, and uncertainty estimation tasks across these modalities. Through rigorous experimental evaluation, the authors demonstrate the benefits of combining structural and temporal context for realistic and reproducible delay prediction.

**Dataset Code Accessibility:**

Yes

**Ethical Considerations:**

No, there are no or only very minor ethics concerns

**Limitations Weaknesses:**

1.	Spatiotemporal Visualization: While the paper includes figures illustrating flight chains and networks, further intuitive visualizations (e.g., delay propagation across a day at a major hub) could improve accessibility for a broader audience.
2.	Graph Modeling Limitations: Although the VGAE + AFM approach outperforms baselines, the graph model architecture remains relatively basic. Future work could explore more expressive GNNs (e.g., heterogenous GNNs or diffusion-based models).
3.	Geographic and Feature Bias: As discussed in Appendix C, the dataset is biased toward North America and lacks fine-grained features such as real-time ATC decisions. These limitations are acknowledged, and extensions (e.g., integrating EUROCONTROL data) would be valuable.

**Strengths Contributions:**

1.	Novel Benchmark Construction: Aeolus is a significant contribution in its own right, bridging the gap between real-world operational complexity and academic benchmarks. The combination of tabular, temporal, and graph-structured data is highly innovative and well-integrated.
2.	High-Quality Engineering: The dataset is meticulously preprocessed, with careful attention to data leakage prevention, temporal slicing, and multimodal alignment. The flight chain and flight network construction are methodologically sound and practically relevant.
3.	Comprehensive Evaluation Protocols: The authors evaluate a broad range of models—from tabular MLPs to LSTMs and VGAEs—on multiple tasks, including regression, classification, and uncertainty estimation (via CRPS), highlighting Aeolus’s flexibility.
4.	Reproducibility and Openness: The authors release their code and data under a permissive license and provide full training details, split strategies, and hardware settings, making this a model of open science.
5.	Impactful Use Case: Flight delay prediction has high societal and economic relevance. The paper connects academic ML with a clear real-world problem, offering a testbed that could inform both research and operational deployment.

---

> ### Author Rebuttal · Authors · 2025-07-26
>
> We sincerely appreciate the reviewers' comments, which are very helpful for our future work. Below are the corresponding responses to each question:
> ### **Q1: Limited Spatiotemporal Visualization**
> **A1:** Thank you for the suggestion. In response to your suggestion for clearer visualizations, we have created two additional visualizations to provide a better understanding of the propagation of flight delays across time and space. These updated visualizations are available in our corresponding GitHub repository.
>
> ### Figure 1. Delay Propagation Visualization in Flight Chains
> This visualization shows the flight chain across major hub airports, with color-coded lines that represent the average arrival delays. It highlights how delays originating from an upstream flight travel through sequential flights in a chain. This visualization is designed to help users better understand the temporal and spatial spread of delays across various locations.
>
> ### Figure 2. Delay Propagation Visualization in Flight Networks
> This second visualization expands on the first by presenting delay propagation within the entire flight network. It shows how delays flow through a network of connected airports, with edges colored according to the intensity of the delay. This approach helps illustrate the system-wide impact of delays, offering a more comprehensive view of how delays propagate across the broader network.
>
> Both visualizations can be accessed in the GitHub repository, where you can explore them further and see how they contribute to a more accessible understanding of flight delay propagation.
>
> We hope these additional visualizations meet your expectations and enhance the clarity of our research for a broader audience.
>
>
> ---
>
> ### **Q2: Basic Graph Modeling — Consider More Expressive GNNs**
> **A2:**We appreciate the reviewer’s valuable comment regarding the limitations of the graph modeling approach. While the VGAE + AFM method outperforms the baseline, we acknowledge that the graph model architecture is still relatively simple. In fact, we considered more advanced models, such as Graph Neural Networks (GNNs), in our early experiments.
>
> It is important to note that the flight network constructed in this study is a graph structure, where flights serve as nodes. Directed edges are formed when there is a spatiotemporal relationship between two flights, indicating a potential delay propagation path. This structure addresses the limitations of flight chains, which cannot capture the delay propagation between different aircraft. However, flight delays are influenced by a range of complex factors, and delay propagation may involve multi-hop nodes. Using GNNs for end-to-end learning faces challenges, such as over-smoothing in deep networks, which can hinder performance.
>
> Moreover, the large scale of the flight network makes directly applying GNNs computationally expensive, and it would be difficult to meet the real-time demands of air traffic control systems. As a solution, we employed Variational Graph Autoencoders (VGAE) to learn the node embedding representations. The encoder part of VGAE uses Graph Convolutional Network (GCN) layers and probabilistic modeling to extract key information from the flight network, using it as auxiliary features. These embedding vectors capture the topological properties and spatiotemporal relationships of flight nodes, such as identifying key connecting flights at hub airports, source and destination flights, or isolated flights, and integrating features from related flights.
>
> Given the potential benefits of more expressive graph model architectures, we recognize the need to explore advanced GNNs in future work. We plan to experiment with heterogeneous GNNs or diffusion-based models and address the challenges outlined, with the aim of improving prediction performance.
>
> ---
>
> ### **Q3: Geographic and Feature Bias**
> **A3:** We thank the reviewer for their thoughtful suggestion. We acknowledge that the Aeolus dataset is indeed geographically biased towards North America. However, we are actively working towards integrating regional datasets, such as those from EUROCONTROL, CAAC, and others, to enhance and diversify our dataset.
>
> Regarding the absence of fine-grained features like real-time ATC decisions, we would like to clarify that, as outlined by [Wandelt et al.,2025], Aeolus already includes five critical feature categories for flight delay prediction: Flight Identification, Time-Related Information, Delay Information, Flight Characteristics, and Weather Data. These features are robust enough to support meaningful research in flight delay prediction and to evaluate the performance of relevant algorithms.
>
> Of course, we also fully recognize that incorporating real-time ATC decisions and similar fine-grained features would significantly enrich the dataset. As noted in Appendix C, we are actively exploring collaborations with airlines or leveraging simulation-based methods to generate such features, further refining the dataset for more precise and comprehensive delay prediction.

---

### Official Review · Reviewer_29AE · 2025-06-30

**Rating:** 5
**Confidence:** 4

**Summary:**

The paper proposes a new flight benchmak, called Aeolus, which extract three aligned modalities from the raw flight operation and weather datasets. The multi-model dataset could fully represent the information in the flight event. The benchmark would push the related research in the area.

**Dataset Code Accessibility:**

Yes

**Dataset Code Comments:**

Dataset and corresponding dataset is available.

**Ethical Comments:**

N.A.

**Ethical Considerations:**

No, there are no or only very minor ethics concerns

**Limitations Weaknesses:**

- All the exsiting methods could not achieve good results in experiments. Does this mean the dataset could not provide enough information between X and Y?
- More advanced methods in tabular could be compared, such as, TabPFN, Catboost.

**Strengths Contributions:**

- This is the first large-scale Multi-modal Flight Delay Dataset designed to advance research on flight delay prediction and support the development of foundation models for tabular data. The major contribution of the benchmark is that the dataset could capture the spatiotemporal dynamics inherent in delay propagation.
- The proposed benchmark has large-scale timeframe, airports, and fights, which could also support many task on it.
- The dataset is also a tabular dataset with large-scale samples.
- The show case also demonstrates the usage of the proposed benchmark.

---

> ### Author Rebuttal · Authors · 2025-07-26
>
> We sincerely thank the reviewer for the detailed and constructive feedback,and we will address each point in turn.
> ### **Q1: Model performance is low — does this indicate the dataset lacks predictive information?**
> **A1:** We respectfully disagree. The observed performance reflects the **high uncertainty and inherent noise** in real-world delay prediction, rather than an absence of predictive signal in the data.
>
> To quantify the relationship between features and delay outcomes, we computed **mutual information (MI)** between key input features and the `ARR_DELAY` / `DEP_DELAY` labels. The results indicate **non-trivial statistical dependencies**, particularly for features related to carrier identity, scheduled time, and airport location:
>
> | Feature            | MI (ARR_DELAY) | MI (DEP_DELAY) |
> |--------------------|----------------|----------------|
> | CRS_ARR_TIME_MIN   | 0.0426         | 0.0638         |
> | OP_CARRIER         | 0.0351         | 0.0687         |
> | CRS_DEP_TIME_MIN   | 0.0414         | 0.0575         |
> | O_LATITUDE         | 0.0405         | 0.0577         |
> | O_LONGITUDE        | 0.0357         | 0.0568         |
>
> These MI values fall within the **typical range (0.01–0.1)** reported for noisy, high-dimensional real-world datasets [Peng et al., 2005; Vergara & Estévez, 2014], and indicate meaningful, though not deterministic, information flow from inputs to targets.
>
> We further verified these associations using **SHAP-based feature importance** from a trained MLP model (see our GitHub repo), which corroborates the relevance of these features.
>
> Additionally, Aeolus is constructed to reflect **naturally occurring, class-imbalanced, and temporally evolving distributions**. In the `ARR_DELAY` task, the ratio of delayed to non-delayed flights is approximately **1:3** (40,902 delayed vs. 130,786 non-delayed), indicating a **significant class imbalance**. This imbalance, coupled with temporal shifts, makes Aeolus **challenging but realistic**.
>
> While many existing models perform well on clean, balanced datasets, we observe substantial performance degradation under **Aeolus’s real-world conditions**, emphasizing the need for benchmarks that support the development of **robust, generalizable models for deployment**.
>
> ---
>
> ### **Q2: Include more advanced methods such as TabPFN and CatBoost**
> **A2:** Thank you for this valuable suggestion. We evaluated several tree-based models, including **CatBoost**, which performed strongly on the `DEP_DELAY` task, surpassing most deep learning baselines while being computationally efficient. This reinforces the view that **traditional tree ensembles remain highly competitive** under high-noise conditions:
>
> | Model     | ARR AUC | ARR ACC | DEP AUC | DEP ACC |
> |-----------|---------|---------|---------|---------|
> | GBDT      | 0.6028  | 0.5830  | 0.6796  | 0.7599  |
> | XGBoost   | 0.6029  | 0.5839  | 0.6809  | 0.7610  |
> | CatBoost  | 0.5992  | 0.5931  | 0.6720  | 0.7599  |
>
> As for **TabPFN**, we attempted to integrate it during early experiments. While **TabPFN v2** performs well on small-scale datasets, it does not scale to Aeolus’s **280,000+ samples**. Even with specialized tools designed to handle large datasets, the preprocessing required by the RandomForestTabPFNClassifier, which is based on Random Forests, still results in excessive GPU memory usage, making it infeasible on standard hardware.We recommend using **subsets of ≤10,000 rows** when experimenting with TabPFN on Aeolus.
> For example, we sampled **3,000 flights** within the period May 10–20, 2020, specifically those scheduled to depart between **12:00 and 13:00**. The model performances on both arrival and departure delay prediction tasks are shown below:
>
> | Model     | ARR AUC | ARR ACC | DEP AUC | DEP ACC |
> |-----------|---------|---------|---------|---------|
> | **TabPFN**| 0.645   | 0.6717  | 0.7161  | 0.8100  |
> | GBDT      | 0.6519  | 0.6600  | 0.6998  | 0.8117  |
> | CatBoost  | 0.6267  | 0.6767  | 0.7211  | 0.8050  |
> | XGBoost   | 0.6135  | 0.6600  | 0.6834  | 0.7933  |

---

> > ### Comment · Reviewer_29AE · 2025-08-06
> >
> > Thank you for the response of the authors. My concerns have been well-solved. I will keep my score.

---

> > > ### Author Response · Authors · 2025-08-06
> > >
> > > Thank you for your constructive comments and insightful questions. Your feedback has provided us with a valuable opportunity to further clarify aspects of our work that were not sufficiently explained in the original manuscript. We sincerely appreciate the time and effort you have dedicated to reviewing our paper.

---

### Official Review · Reviewer_6FE5 · 2025-07-02

**Rating:** 4
**Confidence:** 3

**Summary:**

The authors introduce Aeolus, a large-scale, multimodal benchmark for flight delay prediction that integrates Tabular Features, Flight Chains, and Flight Network Graphs. Aeolus is constructed with strict leakage prevention, temporal splits, and comprehensive preprocessing to support regression, classification, and uncertainty estimation under a unified evaluation protocol.

**Dataset Code Accessibility:**

Yes

**Ethical Considerations:**

No, there are no or only very minor ethics concerns

**Limitations Weaknesses:**

- Absence of critical operational signals such as ATC interventions, crew changes, maintenance events, and passenger itineraries limits causal interpretability and real-world fidelity.
- Predominantly U.S. data (78 % North America) may not generalize to other regions with different regulatory, traffic, or weather patterns.
- The multimodal, large-scale nature demands substantial compute and storage, potentially limiting reproducibility.
- While the paper enforces leakage prevention, it lacks an ablation comparing random vs. temporal splits to quantify leakage impact
- This is only a suggestion: Augment Aeolus with EUROCONTROL, CAAC, or other regional datasets to validate cross-continental generalization and isolate pandemic effects via counterfactual subsets.

**Strengths Contributions:**

- By unifying flat tabular, temporal sequence, and graph-structured data, Aeolus captures the cascading dynamics of delays both within and across aircraft.
- Comprising 9 years of data across 320 airports and 54.7 million flights, Aeolus exposes models to seasonal drifts, COVID-19 anomalies, etc.
- The dataset supports three core tasks: delay regression (MSE/MAE), binary classification (ACC/AUC), and uncertainty quantification (CRPS).

---

> ### Author Rebuttal · Authors · 2025-07-26
>
> We sincerely thank the reviewer for the valuable and insightful suggestions. Below, we systematically address each of the five specific concerns regarding data richness, geographic coverage, scalability, experimental rigor, and broader applicability of Aeolus.
>
> ---
>
> ### **Q1. Missing Key Operational Signals**
>
> **A1:** We agree that incorporating fine-grained operational signals—such as ATC interventions, crew rotations, or maintenance schedules—would enhance causal interpretability and decision support. However, such information is often **unavailable in advance** and typically arises **only shortly before or during flight operations**, making it difficult to incorporate for **pre-departure or booking-time predictions**.
>
> By contrast, the Aeolus dataset is designed to support **early-stage predictive modeling**, where the goal is to provide actionable estimates **well before flight departure**, such as during ticket purchase or initial scheduling. For this setting, we approximate complex operational dynamics through **flight chains** (capturing aircraft continuity) and **flight graphs** (capturing shared resource interference). These structures offer a reasonable proxy for latent constraints under realistic data availability, and this design choice is discussed in **Appendix C**.
>
> We consider extending Aeolus to include delayed operational signals an important avenue for future work, especially for real-time disruption management applications.
>
> ---
>
> ### **Q2. Geographic Bias (U.S.-Centric)**
>
> **A2:** We acknowledge that the current version of Aeolus, built from U.S. BTS data, reflects a **North America–centric distribution** (78.4%). While it spans **320 airports** with diverse spatial and temporal coverage, we agree that broader international inclusion is desirable. To this end, we are actively pursuing integration with **EUROCONTROL**, **CAAC**, and other regional datasets. These efforts will enable more comprehensive benchmarking, facilitate **cross-regional transferability studies**, and better reflect the global nature of air transportation systems.
>
> ---
>
> ### **Q3. Scalability and Reproducibility**
>
> **A3:** We recognize the computational challenges associated with Aeolus, particularly due to its **multimodal**, **temporally granular**, and **large-scale** nature. To promote reproducibility, we have released **full experiment code**, **model configurations**, and detailed **runtime environments**.
>
> Although we do not currently release a lightweight version, we are considering the publication of **modular subsets** based on modality, time window, or geography, which would allow for more accessible experimentation under constrained computational resources.
>
> ---
>
> ### **Q4. Lack of Ablation on Split Strategy**
>
> **A4:** We appreciate this insightful suggestion. To assess the effect of data partitioning, we conducted an ablation study comparing **temporal** versus **random** splits on the `ARR_DELAY` prediction task across seven models. As shown below, **random splits consistently inflate AUC and accuracy metrics**, likely due to information leakage. These results affirm that **temporal splits are critical** for ensuring realistic generalization performance and avoiding overestimation of model capabilities.
>
> | Model         | AUC (Temporal) | ACC (Temporal) | AUC (Random) | ACC (Random) |
> |---------------|----------------|----------------|--------------|--------------|
> | MLP           | 0.600          | 0.689          | 0.645        | 0.688        |
> | AutoInt       | 0.623          | 0.708          | 0.671        | 0.724        |
> | ResNet        | 0.557          | 0.750          | 0.640        | 0.782        |
> | FTTransformer | 0.562          | 0.763          | 0.623        | 0.758        |
> | Tangos        | 0.616          | 0.679          | 0.639        | 0.783        |
> | TabulaRNN     | 0.559          | 0.772          | 0.653    | 0.798    |
> | SAINT         | 0.562          | 0.772          | 0.600        | 0.728        |
>
> We will incorporate this table into the final version to justify our experimental protocol.
>
> ---
>
> ### **Q5. Suggestion — Broader Data and Pandemic Subsets**
>
> **A5:** We appreciate the reviewer’s thoughtful suggestion. We are actively working to expand Aeolus by incorporating **regional datasets** (e.g., EUROCONTROL, CAAC), and we are also considering the construction of **pandemic-specific subsets** (e.g., 2020–2021) to analyze non-stationary patterns induced by large-scale disruptions. These extensions would enable the study of **robustness under distribution shift**, support counterfactual reasoning, and advance understanding of systemic shocks in air transportation networks.

---

### Decision · Program_Chairs · 2025-09-18

**Decision:**

Accept (poster)

**Comment:**

The paper introduces a new large-scale Multi-modal Flight Delay Dataset dedicated to advancing the field of foundational models for tabular data. The dataset addresses the limitations of existing related datasets in capturing spatiotemporal dynamics, by combining three aligned modalities. A diverse set of baseline methods is trained and evaluated on the dataset. The authors added some additional methods during the rebuttal phase. Additional experiments performed to address the reviewers' remarks during the rebuttal phase (localized data,  substantial compute and storage demands, including advanced tabular methods) significantly extend the presented evaluation.

I hereby accept the paper for inclusion in the NeurIPS DB Track,  provided that the final revision incorporates all additional experiments from the rebuttal phase.